# The Mechanism of the Propagation in the Anionic Polymerization of Polystyryllithium in Non-Polar Solvents Elucidated by Density Functional Theory Calculations. A Study of the Negligible Part Played by Dimeric Ion-Pairs under Usual Polymerization Conditions

**DOI:** 10.3390/polym11061022

**Published:** 2019-06-10

**Authors:** Hideo Morita, Marcel Van Beylen

**Affiliations:** 1Honorary Consultant for Synthetic Rubber Division, Asahi Kasei Corp., Torigaoka 50-5, Totsuka-ku, Yokohama 244-0001, Japan; 2Professor Emeritus, Laboratory of Macromolecular and Physical Organic Chemistry, Department of Chemistry, Catholic University of Leuven, Celestijnenlaan 200F, B-3001 Heverlee (Leuven), Belgium; marcel.vanbeylen@kuleuven.be

**Keywords:** anionic polymerization, dimeric ion-pairs, dimeric polystyryllithium, density functional theory, non-polar solvent, polymerization mechanism, styrene

## Abstract

The elementary processes occurring in the anionic polymerization of styrene with dimerically associated polystyryllithium (propagation during the anionic polymerization of dimeric polystyryllithium) in the gas phase and cyclohexane were studied using MX062X/6-31+G(d), a recently developed density functional theory (DFT) method and compared with the polymerization of styrene with non-associated polystyryllithium, which was described in a previous study. The most stable transition state in the reaction of styrene with dimeric polystyryllithium has a structure in which the side chains of styrene and the two chain end units of polystyryllithium are located in the same direction around the Li atom near the reactive site. The relative enthalpy for this transition state in cyclohexane is 28 kJ·mol^−1^, which is much lower than that for the reaction of non-associated polystyryllithium (51 kJ·mol^−1^). However, the relative free energy (which determines the rate constant) for the former is 93 kJ·mol^−1^, which is greater than that for the latter by 7 kJ·mol^−1^, indicating that the latter reaction (reaction with non-associated polystyryllithium) is advantageous over the former (reaction with dimeric polystyrylllithium). Their rates of reaction are also affected by initiator concentrations; in the case of reactions with low initiator concentrations, from which high molecular weight polymers are usually obtained, the rate of reaction corresponding to non-associated polystyryllithium is much larger than that corresponding to dimeric polystyryllithium.

## 1. Introduction

In the case of the anionic polymerization of styrene in non-polar solvents, it is generally accepted that polystyryllitium is mainly associated into dimeric species (PStLi)_2_ in equilibrium with a small amount of non-associated PStLi chains [1,2,3,4,5]. A kinetic order of 0.5 with respect to [PStLi] for this reaction indicates that only non-associated PStLi ion-pairs are able to propagate [6,7]. However, the possibility of a reaction of styrene with dimeric (PStLi)_2_ or higher aggregates was proposed based on experimental data, such as the addition of butadiene to freeze-dried polystyryllithium and the existence of higher aggregates as demonstrated by light and neutron scattering measurements [8,9,10,11,12,13,14]. Counter arguments have been presented against these assertions [15,16,17,18,19]; currently there is no decisive evidence for the advantages of polymerization with dimeric or higher species, when compared to their non-associated counterparts.

In our previous study [20], we described the optimization of the anionic polymerization of styrene with non-associated polystyryllithium using the density functional theory (DFT) calculation method, M062x/6-31+G(d) [21]; the following aspects were observed in that case.
(1)Polystyryllithium was mainly associated into dimeric species and a small amount of non-associated polystyryllithim species reacted with styrene, as expected. The most stable transition state was the one in which Li was located near the phenyl rings of both styrene and polystyryllithium chain end, and the other transition state in which Li was located near the side chains of both styrene and polystyryllithium chain end, which was previously supposed to be the only transition state, was less stable than the former.(2)The penultimate unit effect found by one of the authors [22,23], i.e., slower addition of styrene to polystyryllithium with two or more styrene units compared to polystyryllithium with one styrene unit, was confirmed, and the effect was shown to be caused by coordination of the penultimate styrene units of dimeric polystyryllithium to the Li atoms.(3)The addition of styrene to non-associated polystyryllithium in cyclohexane had the relative enthalpy for the transition state equivalent to the apparent activation energy obtained experimentally by Worsfold et al. [6] and Ohlinger et al. [24].

As described earlier, some researchers are working on the polymerization of styrene with dimeric or higher species. However, it is not clearly shown how and why the dimeric and higher species are more reactive than the non-associated species. The aim of this study is to clarify the mechanism of the anionic polymerization of styrene with dimeric polystyryllithium using DFT calculations in a manner similar to those performed for non-associated polystyryllithium, as described in our previous study [20]; further we intend to compare and contrast the differences between the reactions of the dimeric species and non-associated species.

## 2. Methods

Polystyryllithium species obtained by the addition of styrene to alkyllithium can be denoted as R(St)_m_Li, where R denotes the alkyl group of the initiator. We employed HStLi by setting m = 1 and substituting H for R. Using this structural model, the addition of styrene to dimerically associated polystyryllithium (a propagation reaction) was studied (in our previous study, we set m = 1 and 2 and compared their transition states in order to understand the effect of the penultimate unit and elucidate the reaction mechanism of styrene polymerization with non-associated polystyryllithium). Our purpose in this investigation is to compare the reactions of non-associated and dimerically associated polystyryllitium. Actual calculations on the transition state of St/(HSt_2_Li)_2_ (m = 2) are rather complicated and difficult to perform, therefore we set m = 1 and compared the obtained results of St/(HStLi)_2_ with those of St/HStLi (m = 1) performed in the previous study.

In this study, calculations were performed using the M062x/6-31+G(d)//M062x/6-31+G(d) method [21] with the Gaussian 09W program [25], in essentially the same manner as those performed in our previous study. The dimeric species ((HStLi)_2_) and intermediate complexes (precursor complexes, transition states, etc.) and products of the reaction of (HStLi)_2_ with styrene were optimized, and the obtained geometries, and enthalpy and Gibbs free energy values at 25 °C were used for discussion. The transition states were confirmed to have one imaginary frequency. The precursor complexes and products were obtained by first applying the intrinsic reaction coordinate (IRC) approach [26,27,28] to the transition states and then optimizing the obtained intermediate structures completely. For calculations in cyclohexane, the integral equation formalism (IEFPCM) variant module of the polarizable continuum model (PCM), a widely used method, was employed [29,30].

In order to compare the stabilities of the structures in the gas phase, the values of relative enthalpy in the gas phase (∆*H_r_)* were calculated using the obtained values of enthalpy (*H*) shown in Table A1 of Appendix A.1, according to the following procedure:(a)For (HStLi)_2_,
∆*H_r_* = *H*[(HStLi)_2_] − *H*[(HStLi)_2_]_0_
where *H*[(HStLi)_2_] denotes the enthalpy of a particular (HStLi)_2_, and *H*[(HStLi)_2_]_0_ is the enthalpy of the most stable (HStLi)_2_ that has the lowest free energy among the studied dimers. For the most stable dimer, ∆*H_r_* = 0, and for other dimers, ∆*H_r_* indicates the extent of instability (in a thermodynamic sense) of the particular dimer, (HStLI)_2_, with respect to the most stable dimer.(b)For St/(HStLi)_2_ (precursor complexes, transition states, products, etc.).
∆*H_r_* = *H*[St/(HStLi)_2_] − [*H*(St) + *H*[(HStLi)_2_]_0_]
where *H*[St/(HStLi)_2_] denotes the enthalpy of a particular St/(HStLi)_2_ system and *H*(St) is the enthalpy of styrene. In this case, ∆*H_r_* actually indicates the extent of instability of the particular St/(HStLi)_2_ system with respect to the starting material (styrene and dimeric (HStLi)_2_). ∆*H_r_* for the transition state corresponds to the apparent activation energy of the reaction. 

The values of relative free energy in the gas phase (∆*G_r_*) were also calculated in the same manner as those of ∆*H_r_*, using the values of *G* shown in Table A1. 

∆*H_rch_* and ∆*G_rch_*, the values of relative enthalpy and free energy, respectively, in cyclohexane, were calculated in a manner similar to that used to calculate ∆*H_r_* and ∆*G_r_* in the gas phase, but using the values of *H* and *G* in cyclohexane (Table A2, Appendix A.1). 

In the geometries of all the studied structures, C–Li with distances less than 0.245 nm were marked with full or dotted lines as η-coordinated bonds. 

## 3. Results and Discussion

### 3.1. Addition of Styrene to (HStLi)_2_ in the Gas Phase

**(HStLi)_2_**. To study the addition of styrene to dimeric polystyryllithium, several types of suitable dimeric species should be chosen. To this end, a series of dimeric (HStLi)_2_ used in our previous study on the addition of styrene to non-associated polystyryllithium was employed (Figure 1, **1-a** to **1-f**).
(1)Structures **1-a** to **1-d** exhibited sandwich-type features, while **1-e** and **1-f** were six-membered structures like the four-membered structure of alkyllithium dimers. The former four (HStLi)_2_ with sandwich-type structures have much lower ∆*H_r_* and ∆*G_r_* values than the latter two six-membered structures.(2)In **1-a** and **1-b**, the side chains of the HSt- groups (chain end units) were located near each end of Li−Li one after another, while in **1-c** and **1-d**, they were located near one end of Li−Li. **1-a** and **1-b** exhibited lower ∆*H_r_* and ∆*G_r_* when compared to **1-c** and **1-d**.(3)**1-a** and **1-b** were different with respect to the direction of the side chain of the HSt- group, i.e., in the opposite direction for (**1-a)** and in the same direction for (**1-b)** with respect to the Li−Li line. The ∆*H_r_* as well as ∆*G_r_* values of **1-a** and **1-b** were nearly the same. In the case of **1-c** vs. **1-d** and **1-e** vs. **1-f,** the situation was essentially similar.


**Transition state**. There are several transition states for the addition of styrene to each (HStLi)_2_ shown in Figure 1. These transition states are determined depending on the arrangement of their groups, i.e., whether styrene is coordinated with two Li atoms or only one Li atom, whether the side chain of styrene approaches in the same direction or opposite direction with respect to the side chain of the reacting HSt- group, and from which part of the phenyl group styrene approaches Li away from the reaction site (from C2–C3 or C5–C6 as shown in the drawing in Table 1) if it is coordinated with two Li atoms. In the case of (HStLi)_2_(**1-a)**, which has the lowest ∆*G_r_* value, the possible transition states for the addition of styrene were optimized and the typical transition states are shown in Figure 2 and Table 1 (**2-a** to **2-d**). In each structure, the drawing on the left side represents the side view of the drawing on the right side and the upper drawing is the overhead view of the lower drawing. The carbon atoms of styrene are colored in blue, while those of the reacting HSt- group are brown colored. The blue arrows in each drawing show the main displacement vectors corresponding to the imaginary frequency of the transition state. Accordingly, the blue arrows at the α-carbon of the side chain of the reacting HSt- group and the terminal carbon of the side chain of styrene indicate that these two carbon atoms react in the direction of the arrows to form the product. The relative enthalpies (∆*H_r_*), which are expected to correspond with the apparent activation energies of the reaction, and the relative free energies (∆*G_r_*), which determine the rate constants, of these transition states are also shown in Figure 2 and Table 1. In each of the transition states **2-a** and **2-b**, styrene is coordinated with the two Li atoms, and the styrene side chain approaches from a direction opposite to that of the side chain of the reacting HSt- group. They are different in terms of the portion of styrene coordinating with the Li atom away from the reactive site (C2–C3 or C5–C6), as clearly shown in the left side drawings of **2-a** and **2-b**. In **2-c**, styrene is coordinated with two Li atoms and the side chain of styrene approaches in the same direction as the side chain of the reacting HSt- group, while in **2-d**, styrene is coordinated with only the Li atom near the reactive site and the side chain of styrene approaches in the same direction as that of the reacting HSt- group. Comparing the ∆*H_r_* and ∆*G_r_* values of these transition states, it can be observed that the transition states in which styrene is coordinated with two Li atoms in a direction opposite to that of the reacting HSt- group, i.e., **2-a** and **2-b**, have lower ∆*H_r_* and ∆*G_r_* values than other transition states. In **2-a** and **2-b**, the phenyl group of styrene approaches Li from a different portion and calculations were conducted to determine which of them results in a lower ∆*G_r_* for the transition state.

For the other (HStLi)_2_ moieties shown in Figure 1, the possible transition states were calculated in the same way as for those shown in Figure 2, and the transition states with the lowest ∆*G_r_* for each type of (HStLi)_2_ are shown in Figure 3 and Table 2 as structures **3-b** to **3-f**, along with transition state **2-a** that has the lowest ∆*G_r_* of St/[(HStLi)_2_ (**1-a**)] system. In each of these transition states, styrene is coordinated with two Li atoms and the side chain of styrene is placed in direction opposite to that of the side chain of the reacting HSt- group, which is as expected. The ∆*H_r_* values of these transition states are low, around 22 (for **3-c**) to 39 kJ·mol^−1^ (for **3-e**), while the ∆*G_r_* values are fairly high, around 87 (for **3-c**) to 113 kJ·mol^−1^ (for **3-e**), and their Li−Li distances range from 0.25 to 0.35 nm. From the ∆*G_r_* values of these transition states, transition state **3-c** was found to be the most stable, followed by **3-d, 3-b**, **2-a**, **3-f**, and **3-e**. In **3-c**, the reacting HSt- group is situated oblique to the Li–Li line (red line in the upper drawing of structure **3-c**); the planes of styrene and the unreacting HSt- group are situated parallel to each other (two red lines in the left drawing of structure **3-c**) and the Li−Li distance is 0.35 nm, the longest of all transition states. The orders of magnitude of ∆*H_r_* and ∆*G_r_* of these transition states do not agree with the orders of magnitude of ∆*H_r_* and ∆*G_r_* values of the original (HStLi)_2_. For example, the ∆*H_r_* and ∆*G_r_* values of (HStLi)_2_(**1-c**) and (HStLi)_2_(**1-d**) are higher than those of (HStLi)_2_(**1-a**) and (HStLi)_2_(**1-b**) by about 10 kJ·mol^−1^, as shown in Figure 1. However, the ∆*H_r_* and ∆*G_r_* values of the transition states St/[(HStLi)_2_(**1-c**)] and St/[(HStLi)_2_(**1-d**)], **3-c** and **3-d** in Figure 3, are lower than those of St/[(HStLi)_2_(**1-a**)] and St/[(HStLi)_2_(**1-b**)], **3-a** and **3-b** in Figure 3, by 5−17 kJ·mol^−1^. Further, the ∆*H_r_* and ∆*G_r_* values of the transition states of St/[(HStLi)_2_(**1-e**)] and St/[(HStLi)_2_(**1-f**)], **3-e** and **3-f** in Figure 3, are higher than those of St/[(HStLi)_2_(**1-a**)] and St/[(HStLi)_2_(**1-b**)], **3-a** and **3-b** in Figure 3, by 3−11 kJ·mol^−1^. However, the difference is not as high as that between ((HStLi)_2_(**1-e**) and (HStLi)_2_(**1-f**)) and ((HStLi)_2_(**1-a**) and (HStLi)_2_(**1-b**)), which is about 50 kJ·mol^−1^, as shown in Figure 1.

As described earlier, **3-c** and **3-d** possess lower ∆*H_r_* and ∆*G_r_* than other transition states. In these transition states all three side chains, i.e., those of styrene plus two HSt- groups, are located around the Li atom near the reaction site as clearly shown in **3-g** and **3-h** (a view from another point for **3-c** and **3-d**, respectively). This placement may be responsible for the low ∆*G_r_* of these transition states. In **3-c,** the three said side chains are located in the same direction to Li−Li as shown in **3-g**, while in **3-d**, the side chains of the two HSt- groups interact face to face with each other (**3-h**). The structure of **3-c** may have caused the oblique positioning of the reacting HSt- group with respect to Li−Li, parallel sandwiching of the phenyl groups of styrene and the unreacting HSt- group, resulting in a low ∆*G_r_* value.

**Reaction path (comparison with the reaction of non-associated polystyryllithium)**. The pathway of the reaction system whose transition state is **3-c** is shown in Figure 4; this system will be called system(dim-r) hereafter. The reaction proceeds in three steps. First, an initial complex is formed and it goes through the first and second steps to the final step (the precursor complex, transition state, and product). In Figure 4, only the initial complex and details of the final step are shown because the complete process is complicated and we can discuss the reaction path using the information in Figure 4 (the complete reaction path is shown in Figure A1 of Appendix A.2). It can be observed from Figure 4 that the values of ∆*H_r_* were very low, as (HStLi)_2_ forms the initial complex, precursor complex, transition state, and product without dissociating into non-associated HStLi. The ∆*H_r_* value corresponding to the initial complex was −19 kJ·mol^−1^, indicating an exothermic phenomenon. The ∆*G_r_* values were relatively high, which will be discussed in later sections. The distance between the two carbon atoms participating in the reaction becomes shorter as the reaction proceeds, from 0.37 nm for initial complex **4-a**, through 0.24 nm for transition state **3-c**, to the normal single bond distance for product **4-d** (0.155 nm). 

In this study, the reaction of (HStLi)_2_ (without the penultimate styrene unit) with styrene was used as discussed in the Methods section. Therefore, the reaction of non-associated HStLi (without the penultimate styrene unit) with styrene in the gas phase, which will be called system(mon-r) hereafter, was taken from our original paper and shown as Figure 5. Comparing Figure 4 (system(dim-r)) with Figure 5 (system(mon-r)), it can be noted that the ∆*H_r_* value of transition state **3-c** for system(dim-r) was 22 kJ·mol^−1^, which is much lower than that of **5-b** for system(mon-r), 50 kJ·mol^−1^: this is because (HStLi)_2_ does not undergo any preliminary dissociation in system(dim-r). However, the ∆*G_r_* of **3-c** was 87 kJ·mol^−1^, higher than that of **5-b** by 5 kJ·mol^−1^. As ∆*G* = ∆*H* − T∆*S* (T = absolute temperature and *S* = entropy), this difference is attributed to the difference in the –T∆*S* values of system(dim-r) and system(mon-r). The rate constant of the reaction is related to ∆*G* (which will be discussed in detail in the next section), and the reaction of non-associated polystyryllithium (system(mon-r)) is shown to be advantageous over that of dimer polystyryllithium (system(dim-r)). 

The changes in the ∆*H_r_* and ∆*G_r_* values of system(dim-r) and (mon-r) are schematically shown in Figure 6 and Figure 7, respectively. These figures clearly show that although ∆*H_r_* of the transition state of system(dim-r) related to (HStLi)_2_ is low compared to that of system(mon-r) because of no dissociation in (HStLi)_2_, the ∆*G_r_* value of the former is higher than that of the latter by 5 kJ·mol^−1^, indicating that the route through the latter (system(mon-r)) is the predominant reaction path.

### 3.2. Addition of Styrene to (HStLi)_2_ in Cyclohexane

Anionic polymerization of styrene is generally performed in polar or non-polar solvents. SBR (styrene-butadiene rubber) and styrene-butadiene block copolymers have been produced in non-polar solvents at an industrial scale. Therefore, it is important to study the propagation reaction of anionic polymerization of styrene in non-polar solvents.

The transition states of St/[(HStLi)_2_(**1-c**)] in cyclohexane were optimized using the IEFPCM method of PCM; the transition state with the lowest ∆*G_rch_*, which corresponds to **3-c** in the gas phase and will be called the transition state of system(dim-r) in cyclohexane hereafter, is shown in Figure 8 as **8-a**, along with **8-b** that shows another view of **8-a** from a different aspect. Structure of **8-a** is nearly the same as that of **3-c** (the C−C bond lengths of **8-a** are essentially the same, while the C–Li bonds are larger by a small proportion (up to 0.005 nm), and the Li−Li distance and distance between the two carbon atoms participating in the reaction are almost the same). The ∆*H_rch_* and ∆*G_rch_* values for **8-a** are also shown in Table 3 together with those for St/HStLi (system(mon-r)) that was originally described in our previous paper. A tendency similar to that observed in the gas phase was observed in cyclohexane with respect to the ∆*H_rch_* and ∆*G_rch_* values. The ∆*H_rch_* value of the transition state of system(dim-r) was 28 kJ·mol^−1^, which was much lower than that of system(mon-r), 51 kJ·mol^−1^, owing to no preliminary dissociation of the dimeric species in system(dim-r). However, ∆*G_rch_* for system(dim-r) is 93 kJ·mol^−1^, and higher than that for system(mon-r), 86 kJ·mol^−1^, by 7 kJ·mol^−1^.

The rate of reaction *R* is defined as the product of the rate constant *k* (= e^−∆*Grch*/RT^), styrene concentration [St], and a function of the initiator concentration [Init]. For the reaction of dimeric polystyryllithium, *R*_d_ = *k*_d_[St][Init-d], while that for the reaction of non-associated polystyryllithium is defined as *R*_m_ = *k*_m_[St][Init-m]^1/2^. Therefore,
(1)The rate constant of system(mon-r) is larger than that of system(dim-r).
*k*_m_/*k*_d_ = e^−85,700/RT^/e^−93,300/RT^ = 21(2)Usually, high molecular weight polymers are produced using small amounts of initiator, as can be seen in Table 4, that shows conditions of the experiments preformed to obtain the apparent activation energies for the anionic polymerization of styrene as discussed in our previous study. At an initiator concentration of 10^−3^ mol·L^−1^ which is approximately the average concentration for these experiments, the effect of initiator concentration is [Init]^1/2^/[Init] = 33, and *R*m/*R*d becomes much larger.
*R*_m_/*R*_d_ = (*k*_m_/*k*_d_)([Init-m]^1/2^/[Init-d]) = (21)(33) = 690

Thus, the advantage of the rate constant for the anionic polymerization of non-associated polystyryllithium over dimeric polystyryllithium has been proved, as shown in above (point (1), previous paragraph). Further, to obtain high molecular weight polymers which is often the desired outcome, the reactions are conducted at low catalyst concentrations; under these conditions, the difference in the rate of reaction becomes larger (point (2), previous paragraph). 

Some researchers, especially Fetters et al. [8,9,10,11,12,13] and Watanabe et al. [14], insist that polystyryllithium aggregates higher than dimeric aggregates coexist in the system and that the dimeric and/or higher aggregates participate in the polymerization reaction. In their investigations the existence of small amounts of higher aggregates was demonstrated using light and neutron scattering measurements; in addition, they studied the polymerization of butadiene with freeze-dried polystyryllithium. However, there is no decisive evidence for the advantages of polymerization of dimeric or higher species over that of non-associated species. Frischknecht et al. [32] reported the calculation result that star-like micelles and cylindrical micelles coexist in a polymeric system with butadienyllithium headgroups, based on the experimental results performed by Stellbrink et al. [33]. Calculations were carried out using the classical dipoles of point charges and they neglected the semi-empirical and DFT results of the binding energies shown in Figure 7 to 10 in their paper [32] due to inconsistency with the above Stellbrink et al.’s results, although these semi-empirical and DFT results agreed well with the generally accepted mechanism of the anionic polymerization of styrene [1,2,3,4,5]. 

Our study, including the results described in our previous study, shows that
(1)Although the polymerization of styrene with non-associated polystyryllitnium requires the dissociation of dimeric polystyryllithium for the reaction, its true activation energy for the polymerization reaction is small. Therefore, the polymerization of non-associated polystyryllithium is very rapid. Especially at low catalyst concentrations where high molecular weight polymers are usually obtained, propagation reaction is very powerful.(2)Dimeric polystyryllithium can polymerize styrene. However, it is not as reactive as non-associated polystyryllithium, although its relative enthalpy is lower because there occurrs no preliminary dissociation in the dimeric species.

A reconstruction of their reports, taking our results also into account, is therefore recommended.

## 4. Conclusions

In the case of the anionic polymerization of styrene in non-polar solvents, it is generally accepted that polystyryllitium is mainly associated into dimeric species and only a small amount of non-associated polystyryllithium species can propagate. However, the possibility of the reaction of dimeric polystyryllithium and higher aggregates was proposed by some researchers based on experimental data such as the addition of butadiene to freeze-dried polystyryllithium and the existence of higher aggregates, which was demonstrated using high-performance analytical techniques.

In our previous study, the anionic polymerization of styrene with non-associated polystyryllithium in the gas phase and cyclohexane was studied using M062X/6-31+G(d), a DFT calculation method. It was shown that polystyryllithium mainly associated into dimeric species and a small amount of non-associated species reacted with styrene; its relative enthalpy of transition state in cyclohexane agreed with the apparent activation energies experimentally observed by Worsfold et al. and Ohlinger et al. Further, the most stable transition state was found to be the one with a new structure and the reason for the penultimate unit effect (slower addition of styrene to polystyryllithium with two or more styrene units when compared to that with one styrene unit) was described.

In this study, the polymerization of styrene with dimeric polystyryllithium was optimized in essentially the same manner in which styrene polymerization with non-associated polystyryllithium was optimized in our previous study and the following results were obtained. The most stable transition state of St/(HStLi)_2_ in cyclohexane has a structure in which the side chains of styrene and two HStLi are situated in the same direction around Li near the reactive site (structure **8-a** and **8-b**). Comparing this transition state with the most stable transition state for the reaction of the non-associated polystyryllithium in cyclohexane, it was found that the relative enthalpy for the reaction of the dimeric species was 28 kJ·mol^−1^, which is much lower than that of non-associated polystyryllithium, 51 kJ·mol^−1^; this result is attributed to no preliminary dissociation of dimeric polystyryllithium. However, the relative free energy of the transition state for the reaction of the dimeric polystyryllithium was 93 kJ·mol^−1^, higher than that of non-associated polystyryllithium by 7 kJ·mol^−1^. The reaction rate for this reaction, *R*, is expressed as *k*[St][Init]^1/n^, where *k* is a rate constant expressed as e^−*G*^*^rch^*^/RT^ and [St] and [Init] are the concentrations of styrene and the initiator, respectively, n is 1 for the reaction involving dimeric polystyryllithium and 2 for the reaction involving non-associated polystyryllithium. Therefore, these results demonstrate the advantage of a higher rate constant for the polymerization of styrene with non-associated polystyryllithium when compared to that with dimeric polystyryllithium (*k*_m_/*k*_d_ = 21). At low initiator concentrations where high molecular weight polymers are usually obtained, the effect of initiator concentration can be described using the equation [init-m]^1/2^/[init-d] = 33 at an initiator concentration of 10^−3^ mol·L^−1^ and the difference becomes much larger (*R*_m_/*R*_d_ = (*k*_m_/*k*_d_) ([init-m]^1/2^/[init-d]) = 690).

As described in the preceding section, some researchers proposed that the reaction involving dimeric and/or higher aggregates is highly reactive. However, there is no decisive evidence available on this point. In this study, it was demonstrated that dimeric polystyryllithium can react with styrene, but its reactivity is not as high as that of non-associated polystyryllithium, especially at low initiator concentrations where high molecular weight polymers are generally obtained.

## Figures and Tables

**Figure 1 polymers-11-01022-f001:**
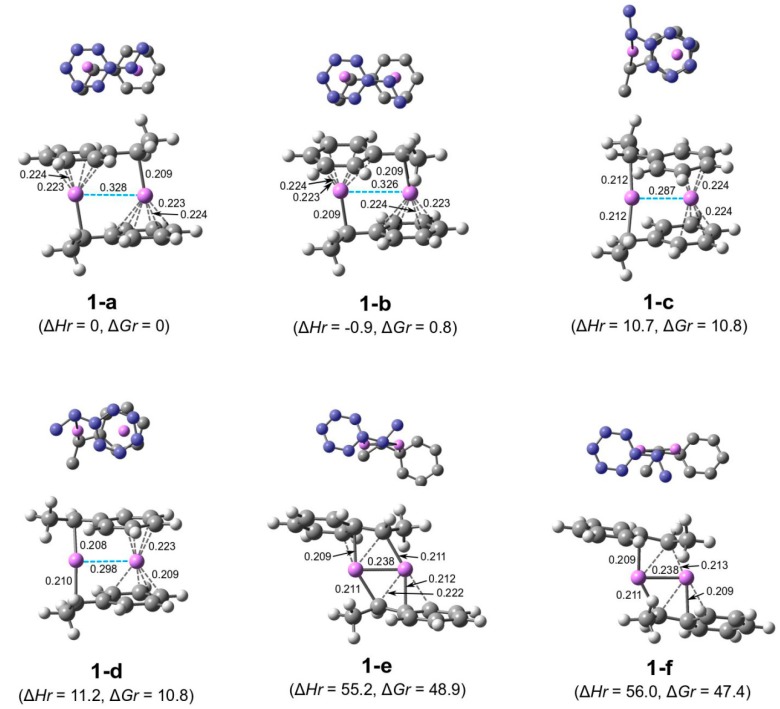
Optimized geometries and relative energies of (HStLi)_2_ in the gas phase originally shown in our previous paper. The small drawing on the top of each structure represents simplified overhead view of the lower drawing; carbon atoms in one of the HSt- groups (the upper HSt- group in the lower drawing) are colored in blue. In the lower drawing, C−Li distances less than 0.225 nm are shown. ∆*H_r_* and ∆*G_r_*, which represent the relative enthalpy and free energy, respectively, are expressed in kJ·mol^−1^.

**Figure 2 polymers-11-01022-f002:**
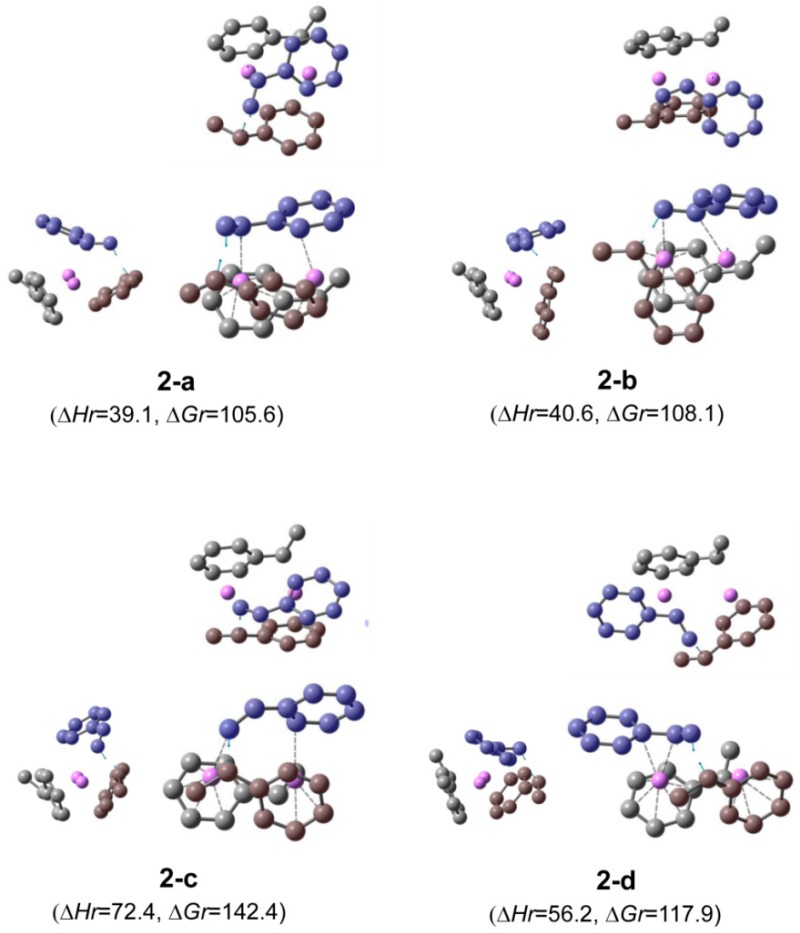
Optimized geometries and relative energies of typical transition states in the gas phase for the addition of styrene to (HStLi)_2_(**1-a**), i.e., (HStLi)_2_ with the lowest ∆*G_r_*, shown in Figure 1. The small drawing on the left side of each structure is the side view of the right-hand side drawing and the drawing on the top is the overhead view of the lower drawing. The carbon atoms of styrene are colored in blue, while those of the reacting HSt- group are colored in brown. The blue arrows in each drawing indicate vectors of imaginary frequency corresponding to each transition state. Detailed arrangements of each structure are shown in the Arrangement column in Table 1 and also in the relevant description in the main text. Hydrogen atoms are not shown. ∆*H_r_* and ∆*G_r_*, the relative enthalpy and free energy, respectively, are expressed in kJ·mol^−1^.

**Figure 3 polymers-11-01022-f003:**
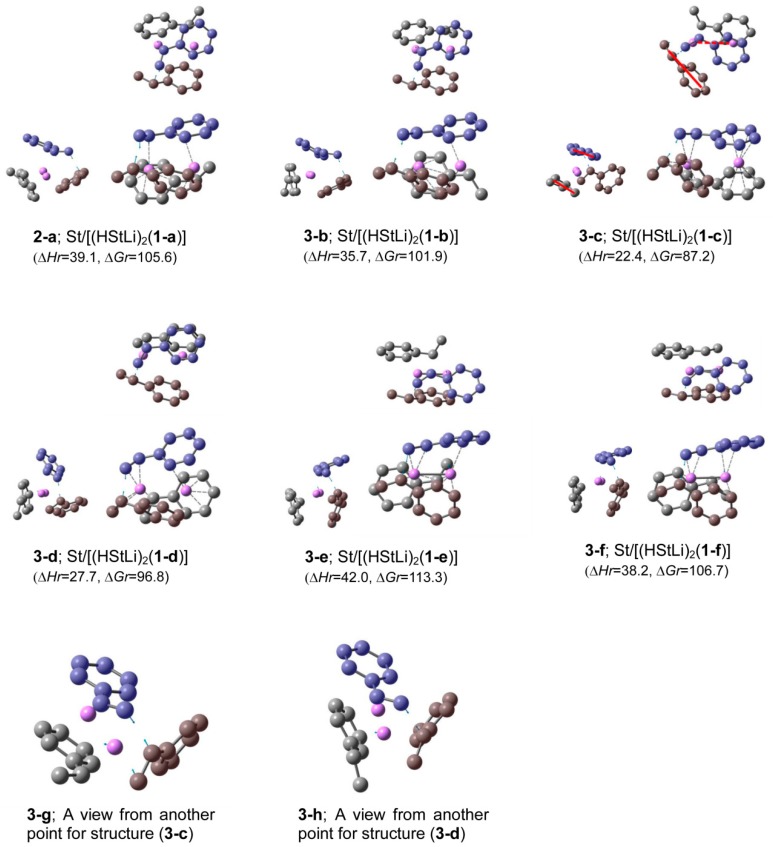
Optimized geometries and relative energies of transition states in the gas phase for the addition of styrene to each (HStLi)_2_ shown in Figure 1. For each (HStLi)_2_, the transition state with the lowest ∆*G_r_*, including **2-a** shown in Figure 2, is shown (**2-a** and **3-b** to **3-f**). Detailed arrangements of each structure are shown in the Arrangement column in Table 2 and also in the relevant description in the main text. **3-g** and **3-h** show a view from another point for **3-c** and **3-d**, respectively. The drawing details can be found in the description for Figure 2.

**Figure 4 polymers-11-01022-f004:**
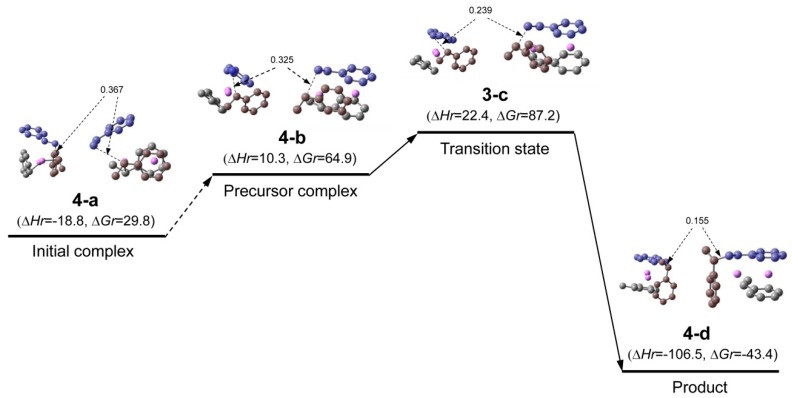
Reaction pathway for St/[(HStLi)_2_(**1-c**)] (system(dim-r)) in the gas phase. Although the reaction proceeds in three steps, only the first complex of the first step (**4-a**) (the initial complex) and the three structures of the final step (**4-b**, **3-c** and **4-d**) are shown here (the complete pathway is shown in Figure A1 of Appendix A.2). The small drawing on the left side of each structure is the side view of the right-hand side drawing. Carbon atoms of styrene are colored in blue, while those of the reacting HSt- group are colored in brown. In each drawing the two carbon atoms participating in the reaction are connected by dotted or full line and the distance between them is shown in nm. Hydrogen atoms are not shown. ∆*H_r_* and ∆*G_r_*, the relative enthalpy and free energy, respectively, are expressed in kJ·mol^−1^.

**Figure 5 polymers-11-01022-f005:**
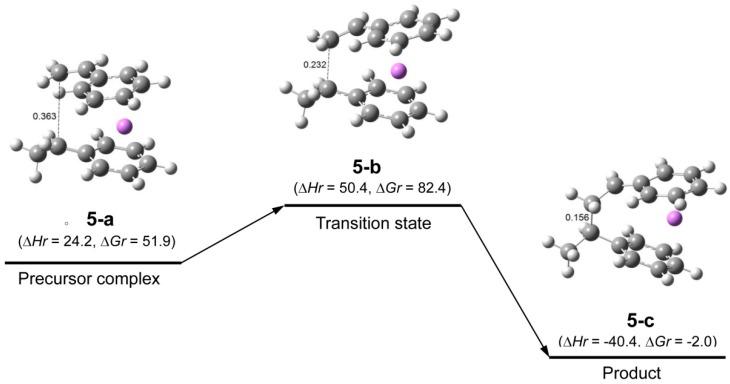
Reaction pathway for St/HStLi (system(mon-r)) in the gas phase originally shown in our previous paper. In each drawing the two carbon atoms participating in the reaction are connected by dotted or full line and the distance between them is shown in nm. ∆*H_r_* and ∆*G_r_*, the relative enthalpy and free energy, respectively, are expressed in kJ·mol^−1^.

**Figure 6 polymers-11-01022-f006:**
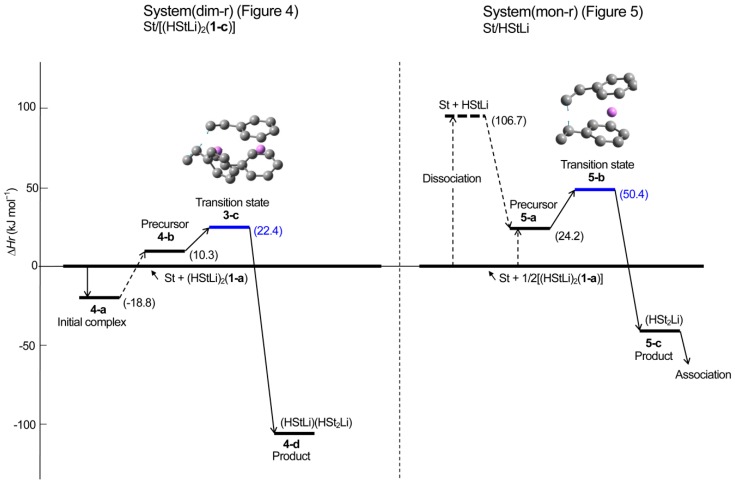
Enthalpy changes in the gas phase for the addition of styrene to (HStLi)_2_(**1-c**) (system(dim-r)) and HStLi (system(mon-r)).

**Figure 7 polymers-11-01022-f007:**
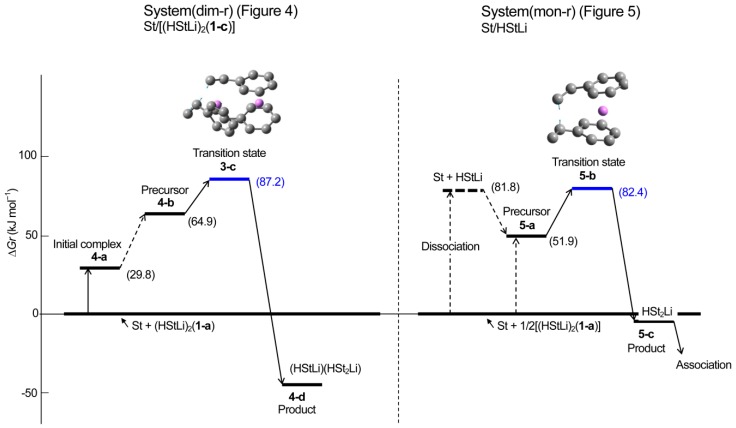
Changes in the free energy in the gas phase for the addition of styrene to (HStLi)_2_(**1-c**) (system(dim-r)) and HStLi (system(mon-r)).

**Figure 8 polymers-11-01022-f008:**
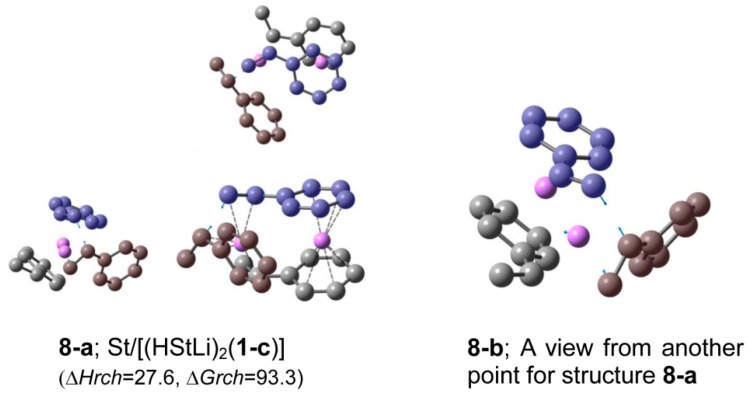
Optimized geometries and relative energies in cyclohexane for the transition state of St/[(HStLi)_2_(**1-c**)] with the lowest ∆*G_rch_*, i.e., **8-a** (whose structure in the gas phase is **3-c** in Figure 3) and **8-b** (another view of **8-a** from a different aspect). The drawing details can be found in the description for Figure 2.

**Table 1 polymers-11-01022-t001:** Arrangement of the transition states shown in Figure 2 for the addition of styrene to (HStLi)_2_(**1−a**).

Item	Detail	2-a	2-b	2-c	2-d
	Coordination of styrene with one Li (Li(1) ^a^) or two Li atoms	Two Li	Two Li	Two Li	One Li
Arrangement	Direction in which the side chain of styrene approaches the reacting HSt- group	Opposite	Opposite	Same	Same
	Portion of the styrene phenyl group ^b^ coordinating with Li(2) ^a^	C2−C3	C5−C6	C2−C3	
	Distance between the reacting C atoms ^c^ (nm)	0.252	0.238	0.206	0.240
Results	Distance between the Li atoms (nm)	0.305	0.289	0.340	0.351
	∆*Gr* of the transition state (kJ·mol^−1^)	105.6	108.1	142.4	117.9

^a^ Li(1) is the Li atom near the reactive site, and Li(2) is the Li atom away from the reactive site.

^b^ Numbering of carbon atoms is shown below.

^c^ Distance between the two carbon atoms participating in the reaction.

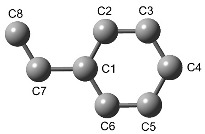

**Table 2 polymers-11-01022-t002:** Arrangement of the transition states shown in Figure 3 for the addition of styrene to each (HStLi)_2_ in Figure 1.

Item	Detail	2-a	3-b	3-c	3-d	3-e	3-f
(HStLi)_2_	Type of (HStLi)_2_	**1-a**	**1-b**	**1-c**	**1-d**	**1-e**	**1-f**
	∆*G_r_* for (HStLi)_2_ (kJ·mol^−1^)	0	0.8	10.8	10.8	48.9	47.4
	Coordination of styrene with one Li (Li(1) ^a^) or two Li atoms	Two Li	Two Li	Two Li	Two Li	Two Li	Two Li
Arrangement	Direction in which the side chain of styrene approaches the reacting HSt- group ^b^	Opp.	Opp.	Opp.	Opp.	Opp.	Opp.
	Portion of the styrene phenyl group ^c^ coordinating with Li(2) ^a^	C2−C3	C2−C3	C5−C6	C2−C3	C5−C6	C5−C6
Results	Distance between the reacting C atoms ^d^ (nm)	0.252	0.248	0.239	0.224	0.227	0.225
	Distance between the Li atoms (nm)	0.305	0.306	0.352	0.318	0.246	0.247
	∆*G_r_* of the transition state (kJ·mol^−1^)	105.6	101.9	87.2	96.8	113.3	106.7

^a^ Li(1) is the Li atom near the reactive site, while Li(2) is the Li atom away from the reactive site.

^b^ Opp.: Opposite.

^c^ Numbering of carbon atoms is shown below.

^d^ Distance between the two carbon atoms participating in the reaction.

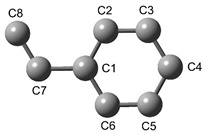

**Table 3 polymers-11-01022-t003:** Relative enthalpies and free energies, respectively, in the gas phase and cyclohexane for the transition states of system(dim-r) and system(mon-r).

Item	In the Gas Phase			In Cyclohexane ^a^		
	Structure	∆*H_r_*	∆*G_r_*	Structure ^b^	∆*H_rch_*	∆*G_rch_*
		kJ·mol^−1^	kJ·mol^−1^		kJ·mol^−1^	kJ·mol^−1^
St/[(HStLi)_2_(**1-c**)] (system(dim-r))	**3-c**	22.4	87.2	**8-a**	27.6	93.3
St/HStLi (system(mon-r))	**5-b**	50.4	82.4	[**5-b**]	51.1	85.7

^a^ The integral equation formalism (IEFPCM) method of the polarizable continuum model (PCM) module was used.

^b^ [**n-x**] represents the structure in cyclohexane whose corresponding structure in the gas phase is **n-x**.

**Table 4 polymers-11-01022-t004:** Experimental conditions and results of the experiments performed to determine the apparent activation energies for the anionic polymerization of styrene (this was also discussed in our previous paper.).

	Experimental Conditions				Results	
	Solvent	Styrene Concentration	Initiator Concentration	Temperature	Rate of Reaction ^a^	Activation Energy ^b^
		mol·L^−1^ × 10^−3^	mol·L^−1^ × 10^−3^	°C		kJ·mol^−1^
Worsfold et al. [6]	Benzene	1.9–28	0.016–39	10–30.3	*k*[St][Init]^0.5^	59.8 ^c^
Ohlinger et al. [24]	Toluene	0.12–15	0.1–15	20–50	*k*[St][Init]^0.5^	59.9
Auguste et al [31]	Ethylbenzene	860–97	0.09–70	25–70	*k*[St][Init]^05^	75 ± 8

^a^*k*: rate constant, [St]: styrene concentration, [Init]: initiator concentration.

^b^ Apparent activation energy.

^c^ The original result was 14.3 kcal·mol^−1^.

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
