# Peer review of "The Mechanism of the Propagation in the Anionic Polymerization of Polystyryllithium in Non-Polar Solvents Elucidated by Density Functional Theory Calculations. A Study of the Negligible Part Played by Dimeric Ion-Pairs under Usual Polymerization Conditions"

_polymers, 2019, doi:10.3390/polym11061022_

Reviewer 1 Report

The authors described a computational study on anionic polymerization of styrene. The results provide new insights into the reaction mechanism.  The results were presented clearly. It is recommended to publish as it is. Only a few minor comments as follows:

1) In the tables, it is necessary to avoid breaking one word or one number into two lines. 

2) Page 6 line 187, the letters and numbers at the bottom of the figure were cut off. 

Author Response

Thank you very much for your excellent review of our manuscript.

You pointed out that there are breakings of one word or one number into two lines in tables and cutoff of the letters and numbers at the bottom of the figure in table 1. These breakings and cutoff were not seen in the original manuscript submitted by us. However when it was modified into the manuscript for reviewers, the breakings and cutoff appeared due to changes in the size of the font and deletion of lines. The newly submitted manuscript is the modified one based on that file and you will be satisfied with the new manuscript.

Reviewer 2 Report

Title: The Mechanism of the Propagation in the Anionic Polymerization of Polystyryllithium in Non-Polar Solvents Elucidated by Density Functional Theory Calculations. A Study of the Negligible Part Played by Dimeric Ion-Pairs under Usual Polymerization Conditions. (Submitted to Polymers)

   This paper describes estimation of the propagation species and the polymerization mechanism in anionic polymerization of styrene in hydrocarbon solvent by DFT calculation. This typed study is very fundamental and important from a scientific viewpoint. This manuscript is clearly written with all necessary results and discussions. However, only 4 out of 33 references are from the year 2000 and onwards. It is hard to convince the reader that the questions tackled are important and timely, if only very dated citations are given.

Basically, I recommend the publication of the manuscript by Polymers, regarding its important information to the field of scientific and industrial polymer chemistry.

Author Response

Thank you very much for the excellent review of our manuscript.

Although we had studied this field for a long time and had known necessary references well, last year we tried to collect all necessary references on association of ion-pairs of styrene and butadiene and their reactivity, by way of precaution, and added some newly found references to our list. Therefore we believe all necessary references, both old and new, are included in our list and are discussed in the manuscript.
